# Real-Life Experience with Ledipasvir/Sofosbuvir for the Treatment of Chronic Hepatitis C Virus Infection with Genotypes 1 and 4 in Children Aged 12 to 17 Years—Results of the POLAC Project

**DOI:** 10.3390/jcm10184176

**Published:** 2021-09-15

**Authors:** Maria Pokorska-Śpiewak, Anna Dobrzeniecka, Małgorzata Aniszewska, Magdalena Marczyńska

**Affiliations:** 1Department of Children’s Infectious Diseases, Regional Hospital of Infectious Diseases in Warsaw, Medical University of Warsaw, Wolska Str. 37, 01-201 Warsaw, Poland; malgorzata.aniszewska@wum.edu.pl (M.A.); magdalena.marczynska@wum.edu.pl (M.M.); 2Department of Pediatric Infectious Diseases, Regional Hospital of Infectious Diseases in Warsaw, 01-201 Warsaw, Poland; adobrzeniecka@zakazny.pl

**Keywords:** children, hepatitis C, ledipasvir/sofosbuvir, real-life, sustained virological response

## Abstract

Background: Available real-world data on the efficacy and safety of ledipasvir/sofosbuvir (LDV/SOF) in pediatric patients are limited. In this prospective, open-label, single-center study, we aimed to present our real-life experience with a fixed dose of LDV/SOF (90/400 mg) for the treatment of chronic hepatitis C (CHC) genotypes 1 and 4 in children aged 12 to 17 years. Methods: We analyzed intention-to-treat (ITT) and per-protocol (PP) rates of sustained virological response (SVR), defined as undetectable HCV viral load at posttreatment week 12, in 37 participants treated with LDV/SOF according to the HCV genotype, baseline liver fibrosis, duration of treatment, and experience of the previous ineffective antiviral treatment. There were 32 patients infected with genotype 1 and 5 with genotype 4. Fourteen (38%) participants were treatment-experienced, two were coinfected with HIV, and three were cirrhotic. Two patients qualified for 24 weeks of therapy, and the remaining 35 received 12 weeks of LDV/SOF treatment. Results: The overall ITT SVR12 rate was 36/37 (97%). One patient was lost to follow-up after week 4 of therapy when his HCV RNA was undetectable. All 36 patients who completed the full protocol achieved SVR (36/36, 100%). PP analyses of SVR12 rates according to the HCV genotype, baseline liver fibrosis, duration of the treatment, and previous ineffective treatment were all 100%. A significant decrease in aminotransferase serum levels was observed in the subsequent weeks of the treatment and at SVR assessment compared to baseline. No serious adverse events were reported. Conclusions: The results of this study confirm previous observations of a suitable efficacy and safety profile of LDV/SOF for the treatment of CHC genotypes 1 and 4 in adolescents.

## 1. Background

It is estimated that over 3.25 million (95% confidence interval 2.07–3.90) children are infected with hepatitis C virus (HCV) globally, which corresponds to a prevalence of 0.13% (0.08–0.16) [1]. Among them, 3500 (2600–4200) subjects are considered to be living in Poland, which makes the HCV prevalence 0.05 (0.04–0.06) [1]. However, according to the data published by the National Institute of Public Health, Warsaw, Poland, between 2010 and 2019, only 545 cases of hepatitis C were reported in patients aged 0–19 years, which suggests that most cases of HCV-infected children remain undiagnosed [2]. Chronic hepatitis C (CHC) in children is usually considered a mild disease with only a slow progression of liver disease. However, recent studies reported a significant proportion of pediatric patients who develop significant fibrosis or even cirrhosis as a result of early infection with HCV [3,4,5]. In addition, Younossi et al. [6] showed that HCV infection in adolescents may be associated with decreased health-related quality of life, poor social functioning, and a reduction in intelligence and memory testing. To prevent these consequences of CHC, early anti-HCV treatment should be implemented. New, extremely effective, and safe interferon-free therapies based on direct-acting antivirals (DAAs) have significantly changed the natural history of CHC, and they have provided a chance for HCV eradication [7]. The first DAA, ledipasvir/sofosbuvir (LDV/SOF), was approved for use in children aged 12–17 years by the European Medical Agency (EMA), Amsterdam, The Netherlands, and U.S. Food and Drug Administration (FDA), Silver Spring, MD, US, in 2017 [8]. Since 2019, LDV/SOF has been used in children aged at least 3 years [9,10]. However, due to the prohibitive prices of DAAs, only a few countries have included recommendations for the treatment of pediatric patients infected with HCV in their national policies and strategies [11,12]. Thus, only a small proportion of children and adolescents with CHC have been treated, mainly during clinical trials. As a result, available real-world data on the efficacy and safety of LDV/SOF in pediatric patients are limited [13,14]. Thus, in this prospective, single-arm, observational, open-label single-center study, we aimed to present our real-life experience with LDV/SOF for the treatment of CHC in children aged 12 to 17 years infected with HCV genotypes 1 and 4.

## 2. Materials and Methods

In Poland, patients below 18 years of age are not included in the national therapeutic programs for CHC. However, courtesy of a donation of LDV/SOF by the pharmaceutical company in August 2019, our single tertiary health care pediatric infectious disease department launched the real-life therapeutic program ‘Treatment of Polish Adolescents with Chronic Hepatitis C Using Direct Acting Antivirals (POLAC project)’. In this project, we qualified consecutive patients aged 12–17 years (weighing at least 35 kg) infected with genotype 1 and 4 HCV for therapy with LDV/SOF (fixed-dose tablet of 90/400 mg). CHC was diagnosed in subjects with over a 6-month duration of disease confirmed with positive nucleic acid testing, HCV RNA, using quantitative real-time polymerase chain reaction (RT-PCR) (Abbott RealTime HCV, Abbott Laboratories, Abbott Park, Illinois, USA; measurement linearity range 12–1.0 × 10 ^8^ IU/mL). Patients were eligible for the treatment regardless of the extent of liver fibrosis or previous ineffective treatment. The duration of treatment was established according to the recommendations of the European Society of Pediatric Gastroenterology, Hepatology and Nutrition (ESPGHAN), Geneva, Switzerland: patients received 12 weeks of therapy unless they were infected with HCV genotype 1 with a history of previous ineffective interferon-based treatment and presented with cirrhosis. This specific group of patients was treated for 24 weeks [15]. Before starting the treatment, the possibility of drug interactions between LDV/SOF and other medicines received by the patient was excluded using the online HEP Drug Interactions Checker provided by the University of Liverpool (www.hep-druginteractions.org).

### 2.1. Treatment Monitoring and Outcomes

All participants in this study were followed every 4 weeks during the treatment, at the end of the therapy, and at week 12 posttreatment. During all visits, physical examination and biochemical evaluation were performed, and adherence to treatment and possible adverse events were analyzed. HCV RNA testing was performed at baseline, at 4 weeks, and at the end of the treatment (EOT). To assess the efficacy of the therapy, a sustained virological response (SVR12) was evaluated based on negative testing for HCV viral load using an RT-PCR method at week 12 posttreatment. Nonresponders were defined as patients with persistent HCV during treatment, and relapsers were considered as cases in which a reappearance of HCV RNA after its previous disappearance during or after the therapy occurred. Biochemical serum testing was performed using commercially available laboratory kits. For both alanine and aspartate aminotransferase (ALT and AST) serum levels, 40 IU/L was considered an upper limit of normal. Liver METAVIR fibrosis was assessed by the FibroScan device (Echosens, Paris, France) [16]. Transient elastography (TE) examination was performed in all patients on the day the patient started treatment, and in patients presenting with significant fibrosis (F ≥ 2), it was also performed at week 12 posttreatment. Body mass index standard deviation (SD) scores (BMI z-scores) were calculated according to the WHO (Geneva, Switzerland) Child Growth Standards and Growth reference data using the WHO anthropometric calculator AnthroPlus v.1.0.4.

### 2.2. Statistical Analysis

Data distribution was evaluated with the Kolmogorov–Smirnov test before elaboration. Qualitative variables were reported as absolute and relative (percentage) frequencies. Quantitative variables were described as medians (interquartile ranges, IQRs), according to their non-parametric distribution. To compare continuous variables between more than two groups, repeated measures analysis of variance (ANOVA) testing was performed. A two-sided *p*-value of <0.05 was considered to indicate significance. All statistical analyses were performed using MedCalc Statistical Software version 20.009 (MedCalc, Ostend, Belgium).

### 2.3. Ethical Statement

The local ethics committee of the Medical University of Warsaw, Poland, approved this study (Number of approval: KB/87/2019; date of approval: 13 May 2019). Written informed consent was collected from all the patients and/or their parents/guardians before their inclusion in the study. The investigation was performed in accordance with the ethical standards in the 1964 Declaration of Helsinki and its later amendments.

## 3. Results

### 3.1. Study Group

Between August 2019 and December 2020, 37 patients qualified for treatment with LDV/SOF. Most of them were infected with genotype 1 HCV (26 with 1b; 4 with 1a; and 2 with undefined 1). Two patients were coinfected with human immunodeficiency virus (HIV) and had received effective antiretroviral treatment. One patient had evidence of previous hepatitis B virus infection (HBV): detectable anti-HBc antibodies with negative HBs antigen testing. Baseline liver stiffness measurement (LSM) revealed significant fibrosis (F ≥ 2 points in METAVIR scale) in 4/37 (11%) patients, including 3/37 (8%) with compensated cirrhosis (Child–Pugh class A). Two of these cirrhotic patients were infected with genotype 1b HCV, and they had a history of previous ineffective treatment with interferon and ribavirin. Thus, they were qualified for 24 weeks of LDV/SOF therapy. The baseline characteristics of the study group are presented in Table 1.

### 3.2. Efficacy of the Treatment

After four weeks of treatment, HCV RNA was undetectable in 31/37 (84%) patients and detectable in 6/37 (16%) patients, ranging between 14 and 942 IU/L (Figure 1). At the EOT, HCV RNA was undetectable in 31/37 (84%) patients, including 4 of the 6 patients with detectable HCV viral load after 4 weeks of therapy. In the remaining 6 cases, the evaluation was not performed due to the ongoing coronavirus disease 2019 (COVID-19) pandemic. Assessment of SVR12 was performed in 36/37 cases; however, in 21 participants, the evaluation of the SVR was postponed from 3 to 12 months as a result of the disruption caused by the COVID-19 pandemic. One patient (infected with genotype 1b, with cirrhosis) was lost to follow-up after week 4 of treatment when his HCV RNA was undetectable. However, home delivery of LDV/SOF was arranged for him, and he completed the 24-week therapy.

The overall intention-to-treat SVR12 rate in this group was 36/37 (97%). All the patients who completed the full protocol and were evaluated at least 12 weeks after the end of treatment achieved SVR12 (36/36, 100%) (Table 2). Intention-to-treat and per-protocol analyses of SVR12 according to the HCV genotype, baseline liver fibrosis, duration of the treatment, and previous ineffective treatment with interferon and ribavirin are presented in Table 2. There were no cases of treatment nonresponse or relapse in our study group.

A significant decrease in both ALT and AST serum levels was observed in the subsequent weeks of the treatment and at SVR assessment compared to baseline (Figure 2A,B).

### 3.3. Tolerability and Safety of the Treatment

All 37 patients received treatment with the oral fixed-dose tablet of LDV/SOF (400/90 mg) once daily, and they all completed the treatment. No patient declared omission of any drug dose or delay in the admission of the drug dose longer than 3 h. The treatment was well tolerated. No serious adverse events were observed in this group. Overall, 11/37 (30%) patients complained of any adverse event, with fatigue as the most common (5/37, 14%). Other observed side effects of the treatment are listed in Table 3. Six patients (16%) suffered from upper respiratory tract infections during the treatment. In addition, two episodes of alcohol intoxication were reported in the study participants receiving treatment.

## 4. Discussion

Our study revealed a 100% efficacy and a suitable safety profile of LDV/SOF treatment in children aged 12 to 17 years infected with genotypes 1 and 4 HCV. This therapy has been approved by the FDA and EMA for use in children aged 3 years and older with CHC based on the results of three open-label single-arm clinical trials [8,9,10]. However, one of the biggest problems of clinical trials is selection bias, which may lead to a mismatch between the trial population and real-world patients. Thus, their results should be confirmed by real-life studies, which would also include specific subgroups of patients, e.g., with liver cirrhosis, HIV/HCV, or HIV/HBV coinfections. In a recently published systematic review with meta-analysis on the efficacy and safety of different DAAs (including LDV/SOF) in children and adolescents with CHC, Indolfi et al. [13] demonstrated that among 39 included studies (both clinical trials and real-life studies) on 1796 subjects, the pooled SVR12 proportion among patients receiving all doses of the therapy was 100% (95% confidence interval 100–100). Among patients who received at least one dose of DAA, the lowest efficacy of the treatment (83%) was reported for children with cirrhosis [13]. However, it should be emphasized that the number of performed studies on LDV/SOF treatment in children and adolescents remains limited, and there is a need for further research in this area. We identified 15 papers (both clinical trials and real-life studies) that analyzed SVR in almost 1000 pediatric patients treated with LDV/SOF (Table 4). In all of these investigations, the treatment was effective in at least 95% of patients, which is consistent with our results, showing SVR in 97% of participants. The few patients who did not achieve SVR were (as in our study) lost to follow-up. There were only single cases described of relapse after the treatment [9]. Pooled data from the 15 abovementioned studies and our investigation on 1016 patients revealed an SVR rate of 98.6% for all genotypes, including 98.4% for patients infected with genotype 1, 75% for genotype 3, and 98.9% for genotype 4 HCV (Table 4). Lower SVR rates for genotype 3 may result from a small number of patients in this group (only 4). It is worth emphasizing that real-life studies on LDV/SOF treatment in children were mainly performed in Egypt; thus, they mainly investigated patients infected with genotype 4 HCV [17,18,19,20,21,22,23]. Studies analyzing the efficacy of LDV/SOF in children infected with genotype 1 are less represented. In a recently published Italian study by Serranti et al. [24], 78 patients were included: 64 infected with genotype 1; 2 with genotype 3; and 12 with genotype 4 HCV. The overall intention-to-treat SVR12 rate was 97.4%, but per-protocol analysis revealed SVR12 rates of 100% overall and separately for all genotypes (1, 3, and 4 HCV). This observation was similar to our results: our per-protocol SVR12 rates were 100% irrespective of the HCV genotype, duration of the treatment, previous treatment experience, or baseline extent of liver fibrosis (Table 2). It is worth emphasizing that the treatment was effective in cirrhotic patients and in two participants coinfected with HIV, as described in detail in another paper [25]. In addition, one of our patients had evidence of past HBV infection with detectable anti-HBc total antibodies. He was closely monitored during and after the treatment, and reactivation of the HBV infection did not occur (ALT and AST levels were normal, HBV DNA was undetectable during and after the treatment) [26]. In a large cohort of adults with HCV/HBV coinfection treated with DAAs, the risk of HBV reactivation in HBsAg-negative/anti-HBc-positive patients was only 0.16% [26]. To avoid HBV reactivation in patients with serologic evidence of a previous or current HBV infection, the clinical and laboratory signs of a hepatitis flare or HBV reactivation should be monitored during treatment with DAAs and posttreatment follow-up. Despite the fact that elevation of the ALT and AST serum levels in patients with CHC is not obligatory and usually not persistent, we found a significant decrease in their levels during the course and after the treatment, which is consistent with observations of the Italian cohort [24].

The treatment with LDV/SOF was well tolerated. No participant discontinued the treatment due to side effects. However, a number of patients complained of the large size of the tablets, which were difficult to swallow. No patient complained of the taste of the drug, which was reported in the cohort of younger children (receiving pellets) [10]. According to the results of the meta-analysis performed by Indolfi et al., the most common adverse events reported in children and adolescents receiving DAAs include headache (19.9%), fatigue/asthenia (13.9%), nausea (8.1%), abdominal pain (7.0%), diarrhea (4.8%), cough (4.0%), and vomiting (2.6%) [13]. Similar side effects were observed in our cohort, with fatigue as the most common (14%). No serious adverse events were reported in the meta-analysis or in our study [13].

Teenagers constitute a special group of pediatric patients; they usually have a sense of immortality, they want to be independent, and their adherence to longer-lasting therapies and obligatory checkups is usually poor. Thus, the value of the study is the fact that it was possible to carry out the entire therapy program and follow-up in 36/37 patients. This indicates that treatment based on DAAs is short and well tolerated by this specific age group of patients.

The treatment duration in our study was established according to the ESPGHAN guidelines, with a minimum duration of 12 weeks [15]. However, there is some evidence based on four reported studies (Table 4) that shortening the duration of LDV/SOF treatment to 8 weeks is equally effective [17,20,24,27]. In the studies by Serranti et al. [24,28], 17 patients in total who were infected with genotype 1 HCV, treatment-naïve, noncirrhotic, and with baseline HCV viral load below 6,000,000 IU/mL were treated with LDF/SOF for 8 weeks. The SVR12 rate in this group was 17/17 (100%). Our data showing that most of the patients had undetectable HCV RNA at 4 weeks of treatment may, to some extent, support shortening LDF/SOF treatment in adolescents.

Our study has some limitations. First, the number of included patients was relatively low. Our study group represents no more than 10% of all pediatric HCV cases diagnosed in Poland during the last decade (2). However, all consecutive patients infected with genotypes 1 and 4 HCV referring to our department were included. To the best of our knowledge, this is the second report on a real-life experience with LDV/SOF in adolescents from Europe, demonstrating the efficacy in participants infected with genotype 1 HCV. As presented in Table 4, studies on the large groups of pediatric patients in this area are unavailable. In addition, our 32 patients infected with genotype 1 HCV represented 10% of all of the pediatric study participants with genotype 1 treated with LDV/SOF (Table 4). Second, gaps in the available data resulting from the disruption caused by the COVID-19 pandemic should be mentioned. However, treatment was completed by all of the patients despite the pandemic, which was achieved thanks to the several efforts that were made to prioritize patient care in our children with CHC, following our own guidelines in this field [31]. In addition, DAA therapies are relatively simple, short, and safe. Thus, less frequent monitoring of patients receiving them might be considered.

In conclusion, the results of this real-life study confirm previous observations based mainly on clinical trials of a suitable efficacy and safety profile of LDV/SOF for the treatment of CHC genotypes 1 and 4 in adolescents, regardless of baseline liver fibrosis or previous ineffective antiviral treatment experience.

## Figures and Tables

**Figure 1 jcm-10-04176-f001:**
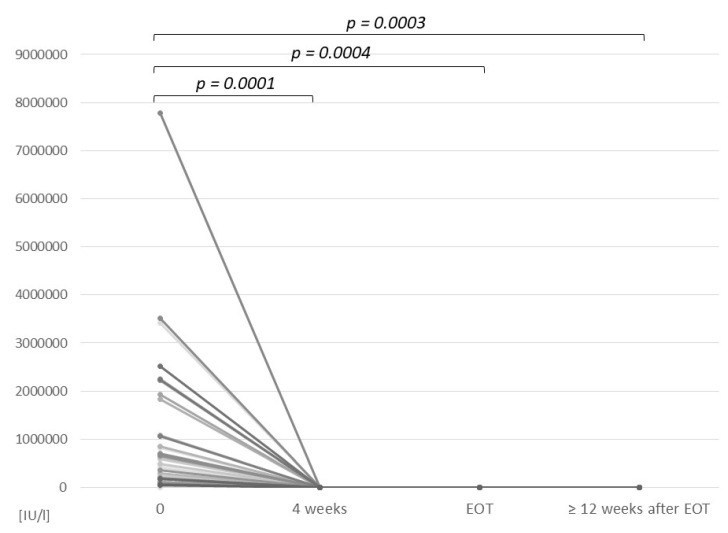
HCV viral load in 37 patients treated with LDV/SOF at baseline, at 4 weeks of treatment, at the end of treatment, and ≥ posttreatment week 12. EOT—end of treatment. Data at EOT were available for 31 patients.

**Figure 2 jcm-10-04176-f002:**
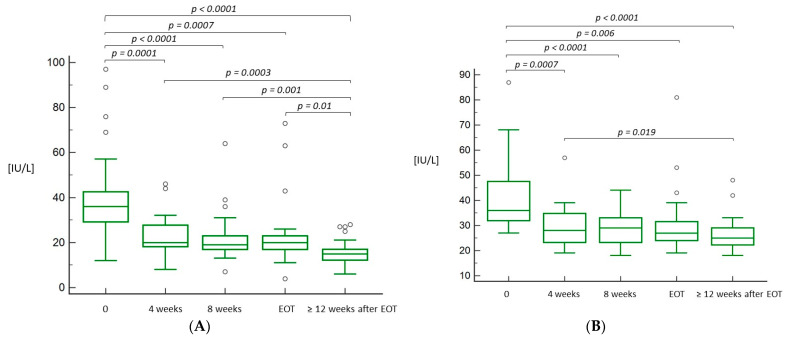
Box-and-whisker plots for alanine aminotransferase (**A**) and aspartate aminotransferase (**B**) levels during and after treatment with LDV/SOF. The top and bottom of each box are the 25th and 75th percentiles. The line through the box is the median, and the error bars are the maximum and minimum. (**A**) 0—start of treatment; EOT—end of treatment. (**B**) 0—start of treatment; EOT—end of treatment.

**Table 1 jcm-10-04176-t001:** Baseline characteristics of 37 patients with chronic HCV infection treated with ledipasvir/sofosbuvir (LDV/SOF).

Characteristics		Number (%) or Median (IQR)
Sex	Male	23 (62)
	Female	14 (38)
Age	Median (IQR)	15 (12; 16)
HCV genotype	1	32 (86)
	4	5 (14)
Mode of infection	Mother-to-child transmission	30 (81)
	Unknown	7 (19)
Previous ineffective treatment with interferon plus ribavirin	Yes	14 (38)
	No	23 (62)
BMI	Median (IQR)	20.4 (17.7; 22.5)
BMI z-score	Median (IQR)	0.23 (−0.65; 0.83)
ALT	IU/mL, median (IQR)	37 (30; 48)
AST	IU/mL, median (IQR)	36 (32; 48)
HCV viral load	IU/mL, median (IQR)	5.83 × 10^5^ (1.8 × 10^5^; 12.6 × 10^5^)
Liver fibrosis (LSM corresponding to METAVIR scale)	F0/F1	33 (89)
	F2	1 (3)
	F3	0
	F4	3 (8)
Anti-HIV	Positive	2 (5)
Anti-HBc total	Positive	1 (3)
Duration of LDV/SOF treatment	12 weeks	35 (95)
	24 weeks	2 (5)

ALT—alanine aminotransferase; AST—aspartate aminotransferase; LSM—liver stiffness measurement.

**Table 2 jcm-10-04176-t002:** Efficacy of LDV/SOF treatment in 37 adolescents with CHC (intention-to-treat and per-protocol analysis).

Patient Characteristics		Number	SVR12 (ITT)	SVR12 (PP)
**Overall**		36/37	97%	100%
HCV genotype	1	31/32	97%	100%
	4	5/5	100%	100%
Baseline liver fibrosis (METAVIR)	F0/1	33/33	100%	100%
	F ≥ 2	3/4	75%	100%
Duration of LDV/SOF treatment	12 weeks	35/35	100%	100%
	24 weeks	1/2	50%	100%
Previous ineffective treatment with interferon and ribavirin	Yes	13/14	93%	100%
	No	23/23	100%	100%

ITT—intention-to-treat; PP—per-protocol analysis; SVR12—sustained virological response.

**Table 3 jcm-10-04176-t003:** Side effects of LDV/SOF treatment in 37 patients.

Symptom	Frequency, Number (%)
Any	11 (30)
Fatigue	5 (14)
Headache	4 (11)
Sleepiness	2 (5)
Diarrhea	2 (5)

**Table 4 jcm-10-04176-t004:** Summary of the studies on LDV/SOF efficacy in pediatric patients with chronic hepatitis C.

No	Patients Age Range (Years)	Number of Participants	HCV Genotype	Duration of Treatment (Weeks)	Number of Patients Achieving SVR12 (%)	Reference
1	12–18	40	4	12	100	El-Karaksy et al., 2018 [19]
2	12–18	46	NA	12	98	Fouad et al., 2020 [27]
3	12–17	100	1	12	98	Balistreri et al., 2017 [8]
4	12–17	144	4	12	99	El-Khayat et al., 2018 [21]
5	12–17	14	1	8	100	Serranti et al., 2019 [28]
6	12–17	78	1, 3, 4	8, 12 or 24	97.4	Serranti et al., 2021 [24]
7	12–17	157	4	8 or 12	98	El-Khayat et al., 2019 [20]
8	12–17	65	4	12	100	Makhlouf et al., 2021 [29]
9	11–17	51	4	12	100	Fouad et al., 2019 [30]
10	9–12	100	4	12	100	El-Araby et al., 2019 [18]
11	6–12	20	4	12	95	El-Shabrawi et al., 2018 [22]
12	6–11	92	1, 3, 4	12 or 24	99	Murray et al., 2018 [9]
13	4–10	30	4	8	100	Behairy et al., 2020 [17]
14	3–6	22	4	8 or 12	100	Kamal et al., 2020 [23]
15	3–5	34	1, 4	12	97	Schwarz et al., 2020 [10]
**Overall and According to the HCV Genotype**						
16	3–18	1016	1, 3, 4	8, 12 or 24	98.6	*
17	3–17	317	1	8, 12 or 24	98.4	**
18	6–17	4	3	24	75	***
19	3–18	649	4	8 or 12	98.9	****

* cumulative data from studies No. 1–4 and 6–15 and from our study (participants of study No. 5 are included in study No. 6); ** cumulative data from the above studies No. 3, 6, 12, 15 and from our study; *** cumulative data from the above studies No. 6 and 12; **** cumulative data from the above studies No. 1, 4, 6–15 and from our study, SVR—sustained virological response.

## Data Availability

The datasets used and analyzed during the current study are available from the corresponding author upon reasonable request.

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
