# Peer review of "Real-Life Experience with Ledipasvir/Sofosbuvir for the Treatment of Chronic Hepatitis C Virus Infection with Genotypes 1 and 4 in Children Aged 12 to 17 Years—Results of the POLAC Project"

_jcm, 2021, doi:10.3390/jcm10184176_

Round 1

Reviewer 1 Report

Pokorska-Śpiewak et al. aimed to describe their real-life experience with SOF/LDV among people aged 12-17. This work has a good purpose, since there is a lack of data regarding HCV treatment in <18 years old people. However, there are some points to address before paper publication.

Materials and Methods

  1. Monitoring of treatment (please, rewrite as ‘Treatment monitoring and outcomes)

Please, use universal descriptions as ‘end of treatment’ (EOT) and ‘sustained virological response at 12 weeks (SVR12). Furthermore, please be clearer when defining the outcomes. For example: ‘non responders were defined as […] Relapsers were defined as […]’. Given the audience, there is no need to describe FibroScan values. You can just state ‘Liver METAVIR fibrosis was assessed by FibroScan®.’

  1. Statistical analysis

This should be slightly rewritten. I suggest: ‘Data distribution was evaluated with Kolgomorov-Smirnov test before elaboration. Qualitative variables were reported as absolute and relative (percentage) frequencies. Quantitative variables were described as medians (interquartile ranges, IQR), according to their non-parametric distribution.

Furthermore, the main problem is the sample size evaluation. You have a known population, with a theoretical prevalence you can extract from national data, and other studies about the use of SOF/LDV among children both for genotype 1 and 4 (e.g. Quintero et al, El-Sayed et al., Balistreri et al., or Indolfi et al.). As consequence, you can calculate your sample size to present a more complete methodology.

  1. Ethical statement

Please, provide the protocol number of your Ethical Committee.

Results

  1. The results are describing too much data, which are also presented in tables. Please, summarize the description presenting only the most frequent features. Then, you will provide the whole data presentation onto tables.
  2. Table 3

Please, delete drug non-related events. The table is showing ‘LDV/SOF side effects’ (also, correct the title with the correct drug acronym). You can discuss other events onto the related paragraph.

Discussion

The discussion should be slightly rewritten. Please, do not start discussion section reporting other studies. First paragraph must report your main findings. Then, comment your results comparing them with available literature.

Other issues

Please, carefully revise your English language. Furthermore, avoid the terms ‘boys and girls’ but use ‘male’ and ‘female’. ‘Patients were qualified for the treatment irrespective of’ (Materials and methods Lines 11-12) should be ‘Patients were eligible for treatment independently/regardless of’

A study limitations section would be appreciated after discussion. Inside that, authors should also acknowledge their limited number of included patients.

In conclusion, the authors’ work is an added value to the current literature in field, but it needs some adjustments before publication.

Reviewer 2 Report

The authors present a real-life study on the ledipasvir/sofosbuvir treatment in adolescents from Poland. This study brings more data to the other studies from Europe and the World regarding DAA results in children with HCV infection.

The paper is, in general, well written with only some issues. One problem could be the timing of the viral load determination (lack of it in some patients at the end of the treatment and the delay after 12 weeks from the end of the treatment). Some improvements should be made.

In the Abstract, there is no sentence about the background, as there is only the aim. The type of the study might be presented in the Methods paragraph. The duration of the treatment should also be presented there and not in the results. 

In Material and methods, the duration of the treatments should be better presented. There is no word about 24 weeks duration. 

In Figure 1, the authors should mention that the data presented are only for 31 patients at the end of treatment. As the viral load is 0 for all the results besides the start of treatment, maybe the figure could be left out from the paper and the data presented in the text.

Also, even though the authors present an important statistical significance of the decrease of transaminases, these parameters are not significant for the follow-up of the disease evolution, as, in the vast majority (probably also in your cohort), these were in the normal range even at the beginning of the treatment. As there are many tables, these figures would be possible to be added as supplementary materials only.
